# Cervical Intraepithelial Neoplasia Grade 3 in a HPV-Vaccinated Patient: A Case Report

**DOI:** 10.3390/medicina58030339

**Published:** 2022-02-23

**Authors:** Mateja Sladič, Pepita Taneska, Branko Cvjetičanin, Mojca Velikonja, Vladimir Smrkolj, Špela Smrkolj

**Affiliations:** 1Faculty of Medicine, University of Ljubljana, 1000 Ljubljana, Slovenia; sladic.mateja@gmail.com (M.S.); vladimir.smrkolj@student.uni-lj.si (V.S.); 2Department of Gynaecology, Division of Gynaecology and Obstetrics, University Medical Centre, 1000 Ljubljana, Slovenia; pepitataneska@yahoo.com (P.T.); branko.cvjeticanin@guest.arnes.si (B.C.); 3Department of Pathology, Division of Gynaecology and Obstetrics, University Medical Centre, 1000 Ljubljana, Slovenia; mojcka.velikonja@gmail.com

**Keywords:** papillomavirus vaccines, Papanicolaou test, cervical intraepithelial neoplasia

## Abstract

Persistent infection with human papillomavirus (HPV) causes almost all cervical precancerous lesions and cancers. Bivalent, quadrivalent, and nonavalent HPV vaccines effectively prevent high-grade cervical intraepithelial neoplasia (CIN3). The effectiveness of HPV vaccination against CIN3 is 97–100% in HPV-naïve populations and 44–61% in the overall population. Although HPV vaccination has substantially reduced the incidence of cervical cancers, several cases of precancerous cervical lesions in HPV-vaccinated patients have been reported. We report the clinical case of a 19-year-old woman whose first Pap smear was diagnosed as a high-grade squamous intraepithelial lesion (HSIL) after quadrivalent HPV vaccination. Colposcopy and cervical biopsy were performed, revealing HSIL/CIN3. Our multidisciplinary team decided to take a conservative approach with follow-up visits with cervical biopsies of this young patient. After six months, spontaneous regression of high-grade cervical dysplasia was observed. Although HPV immunization has shown to be extremely effective in preventing a high proportion of cervical precancerous lesions and cervical cancers, HPV vaccines do not protect against all oncogenic high-risk HPV genotypes. Consequently, healthcare providers must encourage HPV-vaccinated women to still regularly attend national cervical screening programs.

## 1. Introduction

Human papillomavirus (HPV) infection is the most common sexually transmitted infection worldwide. The first infection usually occurs early after sexual initiation [1,2,3]. Persistent HPV infection with oncogenic high-risk HPV (HR-HPVs) genotypes causes almost all cervical precancerous lesions and cancers [4], in addition to nearly 90% of anal [5,6], 70% of vaginal [7], 50% of penile [8,9], 40% of vulvar [10], and 13–72% of oropharyngeal cancers [11,12]. The HR-HPVs genotypes 16 and 18 are responsible for approximately 70% of all cervical cancers, and the HR-HPVs genotypes 31, 33, 45, 52, and 58 cause an additional 20% of cervical cancers. Furthermore, the low-risk HPV genotypes 6 and 11 are responsible for 90% of anogenital warts [13,14].

HPV vaccines were developed to protect against HPV infection by the genotypes that most commonly cause cervical diseases and are routinely provided in numerous countries around the world, including 37 countries in the WHO European Region [15]. Following approval by the European Medicines Agency, the European Union authorized the use of three HPV vaccines: bivalent, quadrivalent, and nonavalent. The European Medicines Agency approved the quadrivalent vaccine (4vHPV; Gardasil^®^/Silgard^®^) in 2006, the bivalent vaccine (2vHPV; Cervarix^®^) in 2007, and the nonavalent vaccine (9vHPV; Gardasil 9^®^) in 2015 [16]. All three vaccines target the HR-HPVs genotypes 16 and 18. Additionally, 4vHPV and 9vHPV also target the low-risk genotypes 6 and 11. Moreover, 9vHPV targets an additional five HR-HPVs genotypes: 31, 33, 45, 52, and 58 [14]. More than 60 countries worldwide, including all members of the European Union, have introduced HPV vaccination in their national vaccination programs. The fully state-funded Slovenian National HPV vaccination program was introduced in 2009/2010 and is recommended for girls attending the sixth grade of primary school. In 2021, the Slovenian cervical cancer screening program and registry ZORA have also made HPV vaccination available to preadolescent boys [17].

Over the last decade, the implementation of HPV vaccination in national vaccination programs worldwide has resulted in a rapid reduction of HPV infection and cervical disease. The efficacy of HPV vaccination is higher in HPV-naïve populations (i.e., without baseline HPV infections) and in countries where HPV vaccination programs achieved timely and high rates of HPV vaccination [18,19,20]. Although HPV vaccination has shown extremely high success in reducing the burden of HPV-related disease, it still does not protect against all known HR-HPVs genotypes that cause cervical cancer. Moreover, HPV vaccination is not effective in clearing an already present HPV infection, genital warts, cervical precancerous lesions, or cervical cancer [21]. Consequently, cervical cancer screening recommendations have not yet been modified for HPV-vaccinated women. Clinicians should advise HPV-vaccinated women to still regularly attend national cervical cancer screening programs [21,22,23].

The aim of this case report is to present a young woman, previously vaccinated with a quadrivalent HPV vaccine, whose Pap smear detected a high-grade squamous intraepithelial lesion (HSIL).

## 2. Case Presentation

We present a 19-year-old woman (Gravida 0) who was vaccinated with 4vHPV (Gardesil^®^) in 2015, at age 15. She first attended the national cervical screening program in June 2019, and her first Pap smear detected a HSIL (Figure 1).

Three months later, in September 2019, a colposcopy was performed, and high-grade cervical dysplasia was suspected. This was followed by a colposcopically directed biopsy. Histopathological examination revealed discrete focus of CIN3/HSIL (Figure 2a, b). p16 immunohistochemical (IHC) staining was performed on a 4 mm tumor section cut from formalin-fixed, paraffin-embedded (FFPE) tissue blocks by using a monoclonal antibody to p16 (CINtec^®^ Histology test) on an automated immunostainer. An HPV-infected specimen with p16 overexpression was used as positive control, and normal tissue without p16 overexpression was used as negative control. The staining result was then evaluated by two pathologists independently and was considered to be positive when more than 70% of tumor cells stained diffusely in both the nucleus and the cytoplasm. Any discrepancy was resolved after review by consensus.

Considering the young age of our patient and her previously received HPV vaccination, our multidisciplinary team, consisting of three gynecological oncologists and one pathologist, decided to take a conservative treatment approach with regular follow-ups. In November 2019, another colposcopy was performed, and high-grade cervical dysplasia was suspected again. Cervical biopsy was repeated and demonstrated chronic cervicitis without high-grade dysplasia. Furthermore, during the same gynecological exam, a clinically validated HPV DNA screening test with genotipization was obtained (Hybrid Capture 2 (HPV DNA Test, Qiagen, Hilden, Germany)) [24]. The result was negative for the HR-HPVs genotypes 16, 18, 31, 33, 35, 39, 45, 51, 52, 56, 58, 59, 66, and 68. Almost four months later, in March 2020, a control colposcopy showed low-grade cervical dysplasia. Colposcopically directed biopsy followed, and the histopathological exam revealed normal squamous epithelium without dysplastic lesions. After six months, in September 2020, the Pap smear detected a low-grade squamous intraepithelial lesion (LSIL).

In accordance with the national guidelines for managing abnormal cervical cancer screening test results, a follow-up gynecological exam with a Pap smear was performed after six months. The Pap smear revealed atypical squamous cells of undetermined significance (ASCUS), and the HPV triage test was negative.

## 3. Discussion

With the introduction of prophylactic HPV vaccination programs worldwide, an evident decline in the incidence of cervical precancerous lesions and cervical cancer has been observed [25]. A large systematic review that included 26 trials showed that HPV vaccination effectively protects against cervical precancerous lesions in adolescent girls and young women aged from 15 to 26 [18]. Another systematic review demonstrated that the introduction of 4vHPV vaccination programs reduced HPV 6, 11, 16, and 18 infections and genital warts by approximately 90%, histologically confirmed LSIL by approximately 45%, and HSIL by approximately 85% [14]. Moreover, a recent population-based cohort study showed a strong correlation between 4vHPV vaccination and reduced incidence of invasive cervical cancer, especially among women vaccinated before the age of 17 [26]. Similarly, in another recent observational study, the authors observed a substantial reduction in cervical cancer and CIN3 incidence in young women after the introduction of the HPV immunization program in England, especially in individuals who were vaccinated between the ages of 12 and 13 years [27].

Our patient had previously been vaccinated with 4vHPV and was later diagnosed with CIN3/HSIL after colposcopically directed biopsy. The strain that caused the HPV infection in our patient was not covered by 4vHPV. Similarly, McLucas et al. presented a case report of CIN3 in a patient following Gardasil® vaccination [23]. These findings emphasize the limitations of vaccinations. Currently, vaccines do not cover 15–20 HPV strains that can cause dysplasia [28]. Additionally, in our study, the pathologist applied IHC staining with p16, which is considered to be a reliable marker of HPV infection with high prognostic prediction. Pathological examination of tissue of our patient presented a diffuse reaction after IHC p16 staining. The latter is highly suggestive of HR-HPVs related lesions, which has been previously presented in several other studies [29]. On the other hand, p16 is considered a surrogate marker for HPV infection. Xu et al. concluded that solitary p16 IHC is insufficient for HPV status detection in patients with oropharyngeal squamous cell carcinoma (OPSCC) with tobacco and/or alcohol exposure but performs better in those without exposure, and additional HPV DNA specific testing may be necessary for accurate HPV status determination [30].

Because of our patient’s young age, her HPV vaccination status, and the fact that she was vaccinated before her first sexual intercourse, our multidisciplinary team envisioned a more likely possibility for spontaneous regression of HSIL and thus decided upon a more conservative treatment approach. Additionally, as the patient was vaccinated before the initiation of sexual activity, it is possible that she contracted the virus after vaccination [31]. This further highlights the rationale for our conservative treatment approach. Following regular colposcopy and Pap smears, spontaneous regression of CIN3/HSIL was indeed observed.

Nevertheless, healthcare professionals should be informed that HPV immunization cannot eliminate an already acquired HPV infection, existing genital warts, or cervical intraepithelial neoplasia. Thus, protection is significantly higher among those vaccinated before their first sexual intercourse and exposure to HPV infection. Additionally, HPV immunization cannot provide protection against all HR-HPVs genotypes, as up to 30% of cervical cancers are a consequence of infection by HPV genotypes not included in HPV vaccines [22,25]. Several HPV types which have been described as oncogenic have not been investigated, e.g., HPV 26, 82, and 53 [32]. Likely, these additional HPV types might have been linked to some extent to the neoplasm identified in the young girl. Moreover, a recent comprehensive review reported that approximately 5.5–11% of all cervical cancers are HPV-negative, which could be associated with truly negative or false-negative results [33]. The truly HPV-negative or HPV-independent cervical cancers are not related to HPV infection and have specific pathogenesis, which consequently leads to little effect of HPV vaccination and testing on their prevention. Conversely, false HPV-negativity could be associated with misclassification of non-cervical cancer, latent HPV infection, non-high risk HPV infection, false-negative HPV testing, or disruption of the targeting fragment. Namely, integration of the HPV genome into the host genome involves disruption of HPV E1, E2, L2, or L1 fragments, which makes HPV testing targeting L1 less reliable than targeting HPV E6 and E7 oncogenes [34,35]. However, other studies demonstrated that a few tumors no longer express HPV E6 and E7 oncogenes during cancer development, which results in an HPV false-negative outcome using HPV E6 and E7 oncogenes mRNA testing [33,36].

Therefore, cervical cancer screening program recommendations have not yet been modified for HPV-vaccinated women, who should be advised to still attend regular Pap smears by their gynecologist at least every three years. The latest issue of the American Cancer Society guidelines still emphasizes the importance of cervical cancer screening programs for all women, regardless of their HPV vaccination status [21].

Lastly, it is important to note that the lack of sensitive assays employed for HPV identification, such as real-time PCR for the viral load and viral mRNA determination and IHC for E6 and E7 protein detection, should be considered a limitation of our case study.

## 4. Conclusions

In conclusion, the optimal approach to screening HPV-vaccinated women has yet to be established. National screening programs and healthcare professionals are obligated to alert individuals about possible precancerous cervical lesions and cervical cancer despite previous HPV vaccination.

## Figures and Tables

**Figure 1 medicina-58-00339-f001:**
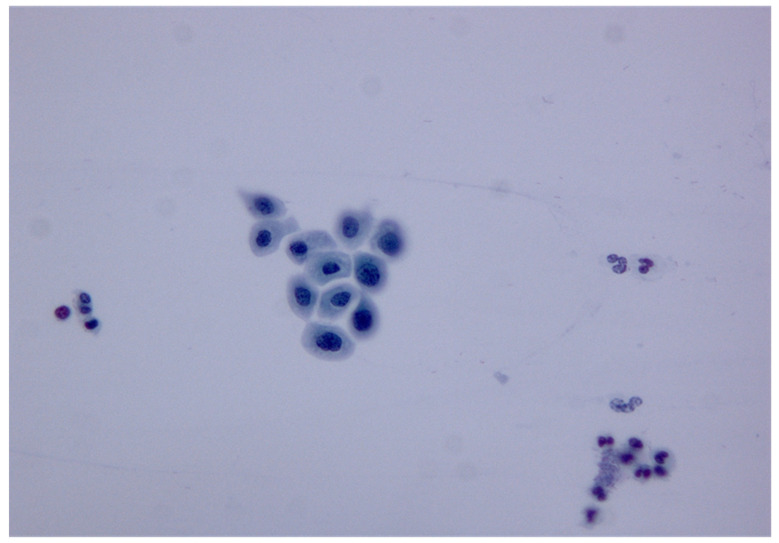
Cytology image of HSIL. Separated cells in a Pap smear presented typical features of high-grade dyskaryosis: slightly enlarged nuclei with hyperchromasia, increased nuclear/cytoplasmic ratios in most cells, and irregularly dispersed chromatin (magnification: 40×).

**Figure 2 medicina-58-00339-f002:**
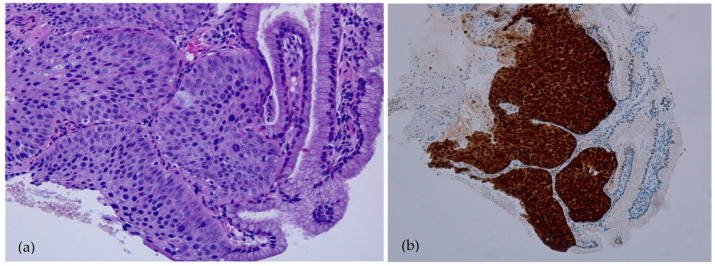
Histopathology image of CIN3/HSIL. The epithelium lacked maturation and consisted of highly atypical cells with hyperchromatic nuclei with increased mitotic activity. The nuclear/cytoplasmic ratio was also increased. (**a**) Hematoxylin and eosin staining (magnification 100×). (**b**) Immunohistochemical staining with p16 revealed a diffuse reaction, which was highly suggestive of CIN3 (magnification 100×).

## Data Availability

The data are available upon request.

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
