# Peer review of "Cervical Intraepithelial Neoplasia Grade 3 in a HPV-Vaccinated Patient: A Case Report"

_medicina, 2022, doi:10.3390/medicina58030339_

Round 1

Reviewer 1 Report

Re: Peer-Review Report

Dear Authors,

          Kindly find the peer-review report for your submitted manuscript (medicina-1594954), for your manuscript by the title: Cervical intraepithelial neoplasia grade 3 in an HPV-vaccinated patient: A case report.

          The article provides novel data on high-grade cervical intra-epithelial neoplasia (CIN) in already HPV-vaccinated young patients. I believe your article is excellent and worthy of publishing at Medicina journal. However, revisions are still needed. You are obliged to correspond and edit the manuscript following the critical appraisal of your article. To complement my peer-review report, I have also uploaded your full-text article, with highlights and additional comments. Please, check and comply with each element.

  • The title is good and representative of the reported case.
  • The abstract is excellent, and it conveys background knowledge and things that are still unknown or of clinical importance concerning HPV-vaccinated women. The case report bears an important clinical message to public health care institutes and healthcare screening programs. Nonetheless, the authors should format the abstract carefully as per Medicina's instructions for authors; available from https://www.mdpi.com/journal/medicina/instructions
  • The authors should avoid reiterating keywords already present in the title of the abstract. Instead, they could rely on indexing terms and keywords, including Embase Emtree terms or PubMed Medical Subject Headings (MeSH); the latter can be accessed via https://www.ncbi.nlm.nih.gov/mesh/
  • The introduction section is reasonable and provides adequate background knowledge, and it debates the rationale and aims of the current case report study.
  • Line #33. The authors should provide an in-text citation for this sentence; orphan statements and sentences providing background knowledge and facts are not allowed throughout the full-text article.
  • Line #48. You already mentioned that HPV is a high-risk genotype, and here you said it's low-risk. Which one is correct?
  • Line #64. Again, an "orphan statement"; an in-text citation is required here.
  • The case presentation section is good. It provides sufficient information and good-quality illustrations, including Pap smear, biopsy, and IHC.
  • The discussion section is excellent, and it provides a good review of the literature, including two systematic reviews, other observational studies, and additional similar case reports. The reported case is of pivotal importance to OBGYN clinical practice, public health, and cervical cancer screening programs.
  • The conclusions section is good, and it conveys a clear take-home message and highlights the study's clinical implications that also possess critical economic dimensions concerning the national screening programs.
  • Line #155. The authors should provide a copy of the consent to the journal's managing editor.
  • The authors should format the bibliographic citations following Medicina's instruction for authors, available from https://www.mdpi.com/journal/medicina/instructions#references. All references are relevant and up to date; however, the authors also relied on data from web pages. They are encouraged to seek relevant information from articles of the highest level of evidence, including randomized clinical trials and systematic reviews. There were no inappropriate self-citations.

Best regards,

The peer-reviewer.

Author Response

We thank the reviewer for his/her valuable insights and constructive comments that have contributed to a better quality of our manuscript. Below are our responses in italics:

Reviewer 1

Dear Authors,

          Kindly find the peer-review report for your submitted manuscript (medicina-1594954), for your manuscript by the title: Cervical intraepithelial neoplasia grade 3 in an HPV-vaccinated patient: A case report.

          The article provides novel data on high-grade cervical intra-epithelial neoplasia (CIN) in already HPV-vaccinated young patients. I believe your article is excellent and worthy of publishing at Medicina journal. However, revisions are still needed. You are obliged to correspond and edit the manuscript following the critical appraisal of your article. To complement my peer-review report, I have also uploaded your full-text article, with highlights and additional comments. Please, check and comply with each element.

  • The title is good and representative of the reported case.
  • The abstract is excellent, and it conveys background knowledge and things that are still unknown or of clinical importance concerning HPV-vaccinated women. The case report bears an important clinical message to public health care institutes and healthcare screening programs. Nonetheless, the authors should format the abstract carefully as per Medicina's instructions for authors; available from https://www.mdpi.com/journal/medicina/instructions

We thank the reviewer for directing us to check the formatting of our article once again, especially the abstract. We have put in minor changes to follow Medicina’s instructions for authors better.

  • The authors should avoid reiterating keywords already present in the title of the abstract. Instead, they could rely on indexing terms and keywords, including Embase Emtree terms or PubMed Medical Subject Headings (MeSH); the latter can be accessed via https://www.ncbi.nlm.nih.gov/mesh/

Following the reviewers comments, we have replaced the keywords with MeSH Headings, we believe that now the article will be more easily discovered.

  • The introduction section is reasonable and provides adequate background knowledge, and it debates the rationale and aims of the current case report study.
  • Line #33. The authors should provide an in-text citation for this sentence; orphan statements and sentences providing background knowledge and facts are not allowed throughout the full-text article.

We have added an appropriate reference for the orphan statement (10.1093/aje/kwf180).

  • Line #48. You already mentioned that HPV is a high-risk genotype, and here you said it's low-risk. Which one is correct?

After careful review we have observed, we made a mistake in the text, and have corrected the mistake. Low risk genotypes are 6 and 11 (not 18 as was written before).

  • Line #64. Again, an "orphan statement"; an in-text citation is required here.

Now we have added an explicit citation, which was before included a few lines lower.

  • The case presentation section is good. It provides sufficient information and good-quality illustrations, including Pap smear, biopsy, and IHC.
  • The discussion section is excellent, and it provides a good review of the literature, including two systematic reviews, other observational studies, and additional similar case reports. The reported case is of pivotal importance to OBGYN clinical practice, public health, and cervical cancer screening programs.
  • The conclusions section is good, and it conveys a clear take-home message and highlights the study's clinical implications that also possess critical economic dimensions concerning the national screening programs.
  • Line #155. The authors should provide a copy of the consent to the journal's managing editor.

We have already sent a copy of the signed mdpi-patient-consent-form upon the editor’s (Assistant Editor, Ms. Juthathip Poofery) request on 31.1.2022.

  • The authors should format the bibliographic citations following Medicina's instruction for authors, available from https://www.mdpi.com/journal/medicina/instructions#references. All references are relevant and up to date; however, the authors also relied on data from web pages. They are encouraged to seek relevant information from articles of the highest level of evidence, including randomized clinical trials and systematic reviews. There were no inappropriate self-citations.

We have rechecked that the citations are formatted according to the Medicina’s instruction. Considering the use of webpages, we have thought carefully and have replaced the WHO statistics webpage with a published article (https://doi.org/10.1016/S2214-109X(16)30099-7), the other webpage is official EMA webpage from where we acquired the information about when were the HPV vaccines approved for use by EMA. We decided to keep this reference.

Reviewer 2 Report

The manuscript entitled “Cervical Intraepithelial Neoplasia Grade 3 in a HPV-Vaccinated Patient: A Case Report” (Manuscript ID medicina-1594954) by Dr. Sladič and colleagues is a case report describing a young 19-old girl with high-grade squamous intraepithelial lesion (HSIL) following quadrivalent HPV vaccination. HPV DNA was not identified in tumor tissues, while FFPEs were positive for p16. Despite, several weaknesses, i.e., lack of methods) I find this submitted manuscript well written in general; the manuscript includes valuable information on HPV vaccination. It provides a clear overview of the effects of HPV vaccination, including the 4-valent vaccine. 

I therefore recommend a major revision. It has some inaccuracies that required rectification

General comments
1. Approximately 5% of cervical cancers can be HPV-negtive (DOI: 10.3389/fonc.2020.606335.) This is an important notion that should be discussed in the papers as supporting the fact that neoplasms might have been arisen from molecular dysregulations independently from HR-HPV E6 and E7 oncoproteins. Similarly, a reduced number of HPV types have been investigated in the tissue.  For instance, several HPV types which have been described as oncogenic (DOI: 10.1097/AOG.0000000000000056,) have not been investigated, e.g., HPV26, 82, 53. Likely, these additional HPV types might have been linked to some extend to the neoplasm identified in the young girl. The authors should discuss this point
2. material and methods section detailing aspects of the molecular analysis and immunohistochemistry should be included. 
3. p16 is considered a surrogate marker for HPV infection (DOI: 10.1016/j.oooo.2019.11.008.) This information should be included

4. The lack of sensitive assays employed for HPV identification, such as real Time PCR for the viral load and viral mRNA determination and IHC for E6 and E7 protein detection, should be considered a limitation of the study. This point should be included in the discussion

Minor
Line 18 Better “cervical cancers”?
Line 33 This sentence is lacking in supporting references. The following supporting references should be included (doi.org/10.3390/vaccines8030473 and PMID: 18701931)
Line 34 I would include “oncogenic” high-risk HPV (HR-HPVs) genotypes. In addition, I suggest substituting high-risk HPV with HR-HPV throughout the text. 
Line 36 references should be included for every HPV-driven quoted. For instance: anal cancer (PMID: 9301544) penile (10.1371/journal.pone.0092208) etc…

Author Response

We thank the reviewer for his/her valuable insights and constructive comments that have contributed to a better quality of our manuscript. Below are our responses in italics:

Reviewer 2

The manuscript entitled “Cervical Intraepithelial Neoplasia Grade 3 in a HPV-Vaccinated Patient: A Case Report” (Manuscript ID medicina-1594954) by Dr. Sladič and colleagues is a case report describing a young 19-old girl with high-grade squamous intraepithelial lesion (HSIL) following quadrivalent HPV vaccination. HPV DNA was not identified in tumor tissues, while FFPEs were positive for p16. Despite, several weaknesses, i.e., lack of methods) I find this submitted manuscript well written in general; the manuscript includes valuable information on HPV vaccination. It provides a clear overview of the effects of HPV vaccination, including the 4-valent vaccine. 

I therefore recommend a major revision. It has some inaccuracies that required rectification

General comments
1. Approximately 5% of cervical cancers can be HPV-negative (DOI: 10.3389/fonc.2020.606335.) This is an important notion that should be discussed in the papers as supporting the fact that neoplasms might have been arisen from molecular dysregulations independently from HR-HPV E6 and E7 oncoproteins. Similarly, a reduced number of HPV types have been investigated in the tissue.  For instance, several HPV types which have been described as oncogenic (DOI: 10.1097/AOG.0000000000000056,) have not been investigated, e.g., HPV26, 82, 53. Likely, these additional HPV types might have been linked to some extend to the neoplasm identified in the young girl. The authors should discuss this point

We thank the author for pointing out important discussion points, which have now been added, together with more references to better support our discussion.

  1. material and methods section detailing aspects of the molecular analysis and immunohistochemistry should be included. 

We have added information about the molecular analysis and immunohistochemical staining where appropriate. Moreover we have added another reference to better support our choice of the methods.

  1. p16 is considered a surrogate marker for HPV infection (DOI: 10.1016/j.oooo.2019.11.008.) This information should be included

As commented by the reviewer we have included the consideration that p16 is a surrogate marker for HPV infection. We have also added the recommended reference.

  1. The lack of sensitive assays employed for HPV identification, such as real Time PCR for the viral load and viral mRNA determination and IHC for E6 and E7 protein detection, should be considered a limitation of the study. This point should be included in the discussion.

We have included the limitation of our study. We now believe this makes our study more rigorous.

Minor
Line 18 Better “cervical cancers”?
Line 33 This sentence is lacking in supporting references. The following supporting references should be included (doi.org/10.3390/vaccines8030473 and PMID: 18701931)
Line 34 I would include “oncogenic” high-risk HPV (HR-HPVs) genotypes. In addition, I suggest substituting high-risk HPV with HR-HPV throughout the text. 
Line 36 references should be included for every HPV-driven quoted. For instance: anal cancer (PMID: 9301544) penile (10.1371/journal.pone.0092208) etc…

We have accepted all minor corrections. Moreover we have added all relevant references where needed.

Round 2

Reviewer 2 Report

The authors have adressed the reviewer concerns. The ms can be acepted in the present form.

The following repetition should be moved before publication:

line 39. "The high-risk HR-HPVs genotypes" should be "The HR-HPV genotypes"